# Reducing the Time-to-Antibiotic by Adapting a Standard of Procedure for the Treatment of Pediatric Cancer Patients Presenting with Febrile Neutropenia—A Comparative Analysis of Two Patient Cohorts

**DOI:** 10.3390/cancers17203280

**Published:** 2025-10-10

**Authors:** Stefano Malvestiti, Brigitte Strahm, Christian Flotho, Markus Hufnagel, Tobias Feuchtinger, Alexander Puzik

**Affiliations:** 1Department of Pediatrics and Adolescent Medicine, Division of Pediatric Hematology and Oncology, Medical Center, Faculty of Medicine, University of Freiburg, 79106 Freiburg, Germany; brigitte.strahm@uniklinik-freiburg.de (B.S.); christian.flotho@uniklinik-freiburg.de (C.F.); tobias.feuchtinger@uniklinik-freiburg.de (T.F.); alexander.puzik@uniklinik-freiburg.de (A.P.); 2Department of General Pediatrics, Adolescent Medicine and Neonatology, Division of Pediatric Rheumatology and Clinical Infectious Disease, Medical Center, Medical Faculty, University of Freiburg, 79106 Freiburg, Germany; markus.hufnagel@uniklinik-freiburg.de

**Keywords:** pediatric oncology, febrile neutropenia, time-to-antibiotic, outpatient

## Abstract

Children undergoing cancer treatment are at high risk of developing febrile neutropenia, a potentially life-threatening condition. Rapid administration of antibiotics is essential, and the time-to-antibiotic (TTA) is considered a key measure of care quality. In this retrospective study, we evaluated whether simple modifications to a standard operating procedure (SOP), such as defining a clear 30 min target and ensuring immediate access to antibiotics, could improve treatment delivery. After SOP implementation, the TTA was reduced by more than half, and a significantly higher proportion of patients received antibiotics within the target time window. While complication rates remained unchanged in this cohort, shorter treatment times represent an important improvement in quality of care. These findings underline the importance of operational measures in optimizing care quality and provide a feasible strategy to improve the management of febrile neutropenia.

## 1. Introduction

Febrile neutropenia (FN) is a common complication in pediatric patients receiving chemotherapy. A bacterial bloodstream infection is identified in 10–25% of the patients and the possibility to develop sepsis and septic shock is estimated to be 20–30% and 5–10%, respectively [1]. Despite continuous efforts to improve the treatment of FN, this complication still ranks among the most common causes of death in patients with cancer. The mortality rate may reach approximately 3% in pediatric and 10% in adult cohorts [2,3,4]. FN remains a medical emergency and requires an immediate clinical response.

The standard of care consists of the rapid identification of at-risk patients upon presentation in the emergency department to initiate appropriate diagnostic and speedy therapeutic measures. The prompt administration of broad-spectrum antibiotics within one hour after admission significantly reduced the mortality in cancer patients [3,5]. The interval between admission and antibiotic administration, also referred to as time-to-antibiotic (TTA), has been therefore used as a quality-of-care (QOC) indicator in the treatment of patients with FN. In line with that, many oncology centers developed empirical strategies aiming to reduce antibiotic timing in patients presenting with FN [6,7]. Keng et al. introduced a febrile neutropenia pathway (FNP) in the Cleveland Clinic (OH, USA) to tackle previously detected causes for delay of the antibiotic administration. The Cleveland FNP included (a) the classification of FN as a potentially lethal condition to the same urgency level as myocardial infarction, (b) the introduction of preferential triage procedures for FN patients in the emergency department and (c) the creation of a standardized order set in the electronic medical chart to reduce treatment variability. Those initiatives significantly reduced median TTA in that adult medical center from 169 min to 81 min [8].

Based on the results in adult cohorts, TTA has become a QOC measure in pediatric patients. However, the body of evidence in the pediatric setting is much weaker and the prognostic role of TTA on adverse events (AEs) such admission to intensive care unit (ICU), development of sepsis, shock and mortality is still controversial [6,7,9]. Rosa et al. showed a reduction in negative outcomes in a mixed cohort of patients (pediatric and adult) when the first antibiotic administration occurred within 30 min of presentation [3]. Salstrom et al. demonstrated that TTA < 60 min correlates with a reduced rate of admission to ICU [10]. Conversely, Koenig et al. observed improvement in outcome by a reduced TTA only in pediatric cancer patients with FN and severe disease (defined as “severe sepsis” or “reduced clinical condition”) [9].

Hence, we analyzed in the present study a simple measure to reduce TTA in pediatric cancer patients. The aim of this investigation was to determine whether an adaption of a local standard of procedure (SOP) for pediatric patients with FN could expedite TTA, trying to provide more insight into the value of short TTA and in its impact on the outcome.

## 2. Materials and Methods

### 2.1. Study Design

This is a retrospective, single-center, cohort study with the goal to assess the efficacy of a simple adaption of a standard of procedure (SOP) that endorses the rapid administration of broad-spectrum antibiotics to pediatric cancer patients with FN. Inclusion and exclusion criteria were defined a priori. We included children with cancer presenting with FN to the pediatric emergency department (pedER) or to the pediatric oncology outpatient department (OD) of a large university medical center in Germany over a period of two years. Sample size was therefore determined by the timeframe. The total number of pediatric cancer patients admitted to our medical center is 80 to 100 per year. Based on the date of adaption of the SOP (15 May 2020), enrolled patients were divided into two cohorts—one year before (pre-SOP, i.e., 1 May 2019–14 May 2020) and one year after (post-SOP, i.e., 15 May 2020–31 May 2021) adaption of the SOP. Inclusion criteria comprised informed consent (by legal caregiver), age < 18 years at time of diagnosis of cancer, fever + neutropenia or afebrile neutropenia + impaired general condition or pathological vital signs (see “Definitions” below). Criteria for exclusion were the absence of any of the inclusion criteria. All patients included in this study had an indwelling central venous line as a hospital standard of procedure for pediatric oncology patients and did not receive any antibiotic prophylaxis, except for pneumocystis jirovecii prophylaxis.

The primary endpoint (PE) consisted of determining the median TTA before and after adjustment of the SOP and assessing the number of FN episodes in which target TTA was met. The secondary endpoint (SE) of this study was to evaluate the occurrence of different adverse events (AEs) over the course of FN episodes within the two cohorts, i.e., (1) length of hospital stay (LOS), (2) sepsis, (3) admission to ICU, (4) administration of vasopressors, (5) need of respiratory support, and (6) mortality. Moreover, the authors hypothesized that the improvement in TTA was sustainable over the post-SOP period of the study.

### 2.2. Local SOP for FN

Based on a previous SOP for the treatment of FN in children with cancer, a procedural algorithm was developed and approved collaboratively by the Department of Pediatric Oncology and the Division of Infectious Diseases. Design followed current German guidelines for the treatment of FN and including local bacterial resistance patterns [4]. After adjustment, the SOP was reviewed, presented and discussed during several routine meetings of the pediatric hospital staff. The hospital staff also received email alerts and training sessions. Additional details regarding the SOP implementation are provided as Appendix A.

Compared to the previous SOP, according to which antibiotics should be administered as soon as possible, the novel one included a clear timeframe of maximally 30 min between admission to the pedER or OD and start of the administration of antibiotics, even without waiting for the absolute neutrophil count (ANC). The TTA was set as low as (logistically) possible to improve our internal processes and motivate staff to speed them up. In addition, a dose of the already reconstituted and routinely applied broad-spectrum antibiotic (piperacillin-toazobactam) was stored around the clock on the pediatric oncology ward. No other major changes were made and no relevant changes in concomitant variables (like staff, buildings) occurred. Figure 1A represents a schematic overview of the new procedural algorithm. A thorough explanation of the SOP algorithm is available as Appendix A.

Minor adjustments to the antibiotic treatment were foreseen in the SOP. In case of volume-responsive hypotension or suspected abdominal infection (e.g., severe mucositis) the initial antibiotic therapy had to be complemented with an aminoglycoside (tobramycin). Addition of glycopeptide antibiotics (vancomycin) was indicated if patients’ symptoms suggested a cutaneous or a central-venous-catheter associated infection, or in patients with ALL or AML after high-dose cytarabine. In case of sepsis or septic shock, broad-spectrum reserve antibiotics (meropenem and vancomycin) were immediately administered. Patients colonized with multidrug-resistant bacteria received antibiotics based on their individual resistance profile.

### 2.3. Definitions

Fever = temperature either > 38 °C (100.4 °F) in repetitive measurements for >one hour, or ≥38.5 °C (101.3 °F) in a single measurement.Granulocytopenia = ANC < 0.5 × 10^9^/L (<500/μL) on day of admission or a falling trend with expected ANC < 500/μL within the following 48 h (i.e., imminent neutropenia). The terms neutropenia and granulocytopenia are synonyms within this manuscript.Febrile neutropenia (FN) = simultaneous presence of fever (1) and neutropenia (2). In our institution, we treat pediatric cancer patients presenting with afebrile neutropenia (or imminent neutropenia) and impaired general condition (e.g., severe fatigue, abdominal pain refractory to non-opioid analgesics, reduced level of consciousness) and/or vital parameters indicating presence of pediatric Systemic Inflammatory Response Syndrome (SIRS: hypotension, tachycardia, hypoxia [11]) as patients with FN according to the corresponding German guideline [4]. Afebrile patients were therefore included in this study to reflect a real-world cohort of FN patients.Length of Stay (LOS) = time interval between the admission and the discharge of the patient.Sepsis and shock were defined according to the International pediatric sepsis consensus conference [11].Time-to-antibiotic (TTA) = lag of time between hospital admission and start of the administration of antibiotics.Composite adverse events = presence of ≥ one of the following adverse events: sepsis, admission to ICU, administration of vasopressors, need of respiratory support and death.

### 2.4. Data Collection

Data were collected from electronic patient records (EPRs) and available paper-based documentation (e.g., doctors’ letters and lab reports) of all children and adolescents with cancer admitted to the University Children’s Hospital during the study period. Their EPRs were systematically screened for the German Diagnosis-related Groups codes for fever, neutropenia and/or FN. Next, episodes of FN occurring during an inpatient stay were excluded. Data gathering included individual documentation of vital signs on admission (i.e., temperature, heart rate, respiratory rate, blood pressure, oxygen saturation), diagnostic workup (lab values, bacterial cultures, imaging results), treatment decisions and applied therapy (antibiotics, vasopressors or IV fluid bolus, respiratory support) and treatment outcome (LOS, transfer to ICU, sepsis, death). The EPRs further provided the time of arrival in hospital, time of administration of antibiotics and time of discharge.

### 2.5. Statistical Analysis

Data are reported as mean, median, standard deviation, confidence intervals (CIs) interquartile range (IQR) and frequency (%). For descriptive statistics Chi-square test, unpaired t-test and one-way ANOVA were performed. The statistical analysis was performed using Graphpad Prism^®^ (Version 10.1.2) and Social Science Statistics^®^ (V22.0) by an independent person.

## 3. Results

### 3.1. General Characteristics and Selection of FN Episodes

In total, the screening process identified 286 episodes of FN in 117 patients during the study period. A total of 21 episodes were excluded, because they did not fully meet inclusion criteria. Of the remaining 265 FN episodes, 127 occurred before the adaption of the SOP and 159 episodes thereafter. Furthermore, in 38 cases (22 before and 16 after SOP adjustment) no FN as defined in this study occurred, as patients did have fever, but either no severe neutropenia manifested, or fever was a bystander of another defined condition. These episodes were included in the analysis of the TTA, since patients were treated as having FN. However, we refrained from including them in the analysis of the secondary endpoints.

Patient characteristics are listed in Table 1. Since there is no data indicating an influence of gender on TTA or infectious adverse events, the only difference between groups in gender ratio (pre-SOP, M:F = 6:4 vs. post-SOP, M:F = 4:6; *p* < 0.01) were considered irrelevant.

### 3.2. Reduction in TTA After Adaption of the SOP

Overall, median TTA significantly declined from 93 min (pre-SOP) to 44 min in the post-SOP period (N = 265, * *p* < 0.001, Figure 2A). Despite the 5.6-fold increase in FN episodes with antibiotic administration within 30 min (pre-SOP 5.6%, post-SOP 32.9%, *p* < 0.0001), the target TTA was still not met in > 50% of the FN episodes (Table 2). The adaption of the SOP appeared to impact more significantly TTA in the pedER (* *p* < 0.001, Figure 2B). Here, the FN episodes treated within 30 min increased from 4.3% to 36.1% (8.4-fold) and between 31 and 60 min from 14.0% to 35.2% (2.5-fold). Moreover, the observed improvement in TTA persisted over time during the study period (Figure 2C).

### 3.3. Shorter TTA in Patients with Signs of SIRS or High Fever

Patients who displayed tachycardia [12] had a shorter median TTA (52 min vs. 75 min, *p* < 0.01) and the degree of fever was linked to a more rapid administration of antibiotics. More precisely, higher temperature negatively correlated with TTA (Pearson’s correlation score −0.18, *p* < 0.01). No significant relationship was identified between arterial blood pressure and TTA (Table 3).

### 3.4. TTA and Adverse Events

There was no significant difference before and after the adaption of the SOP regarding LOS, sepsis rate, rate of admission in ICU, administration of vasopressors, and need of respiratory support or a composite of all AEs (Table 4). One patient died in the pre-SOP cohort: a female, 13-year-old patient with ALL experiencing sepsis in need of respiratory and circulatory support, who received the antibiotic 256 min after presentation. Post-SOP, no patient died from the complications of an FN episode or in temporal connection with it.

A subgroup analysis investigated the relationship between TTA and outcomes per disease groups, but analysis was limited by low numbers of FN episodes and AEs. The rate of manifest sepsis in hematologic malignancies was significantly higher than in solid tumors (Appendix A). The frequency of longer LOS (especially LOS > 9 days) was also higher in hematologic malignancies (46.5% vs. 10%, *p* < 0.0001). Despite a significant reduction in TTA in both disease groups, no statistically significant difference in the occurrence of AEs could be displayed (Appendix A).

### 3.5. Association Between Laboratory Values and Clinical Outcomes

The initial platelet count and highest CRP level during the hospital stay showed a correlation with the LOS (Appendix A). Additionally, a thrombocyte count < 50 × 10^3^/µL or a maximum CRP level > 90 mg/L correlated with a higher risk of developing AEs (respective RRs and 95%-CI are listed in Appendix A).

## 4. Discussion

This study investigated the impact of a simple SOP adaptation on shortening the TTA in pediatric cancer patients presenting with FN. By defining a concrete target TTA (≤30 min) instead of “as soon as possible”, ensuring the availability of pre-dissolved antibiotics, and providing staff training, the median TTA was reduced from 93 min to 44 min—a >50% decrease, comparable to or even exceeding improvements reported in similar studies [7,8,13]. Despite this progress, the primary benchmark of ≤30 min was achieved in only 32.9% of FN episodes, meaning that the majority of patients still received antibiotics later than intended. Nonetheless, the proportion of patients treated within the 30 min window increased more than five-fold compared to the pre-SOP cohort, highlighting a meaningful, though incomplete, step forward. Evidence directly supporting the benefit of a TTA < 30 min in pediatric oncology patients is scarce [3], whereas stronger consensus exists regarding the critical importance of antibiotic administration within 60 min, often referred to as the “golden hour” of FN [4,9,10]. In this respect, our intervention led to a substantial improvement: two-thirds of FN episodes were managed within 60 min, compared to only 22.7% before SOP implementation. Nevertheless, since the majority of patients still failed to receive antibiotics within the predefined 30 min target and a third of patients failed to receive them within 60 min, the intervention cannot yet be considered sufficient. Additional efforts will therefore be required to further optimize the timeliness of FN management discussed below.

Interestingly, even though we observed a trend to a shorter TTA in the OD in the post-SOP cohort, a statistically significant difference was determined in the pedER group only. The median TTA in the pre-SOP cohort was already lower; however, not significantly, for patients presenting to the OD. The number of FN episodes cared for at the OD was lower compared to the pedER (N = 64 vs. N = 201), making the two groups less comparable. Moreover, logistic and resource differences might have played a role in the different effects. The pedER was equipped with more rooms and personnel and an expert triage nurse permitted a timely first encounter and recognition of the emergency. In the OD, consultation hours are prescheduled and typically fully booked weeks in advance. As a result, patients presenting with FN must be accommodated in between preexisting appointments, adding to the daily workload. This often leads to delays in the initial management of these patients, as consultation rooms remain occupied by other patients and personnel and physical resources are already allocated to other cases. In contrast, the pedER exclusively manages emergency cases. Furthermore, the limited exchange of nursing staff between the OD and the pedER (see “SOP implementation” in the Appendix A) may contribute to less frequent exposure to FN episodes in the OD compared to the pedER. Finally, social and psychological reasons that lie within human nature and are difficult to quantify might have led to a less timely reaction from the side of the caregivers as well as the medical staff. A typical example would be caregivers that do not inform the medical staff that their child has developed fever before medical encounter, because to them the child had not appeared impaired yet, and wait for their scheduled appointment in the OD to share this precious information. The difference seen between pedER and OD draws attention to the need to involve all stakeholders when adapting SOPs.

Reflecting on possible reasons for a delay in TTA, we analyzed different time windows during the day linked to available personnel. The time of presentation during day (8 a.m.–4 p.m.), evening (4 p.m.–0 a.m.) and night (0 a.m.–8 a.m.) was not related to the TTA, both in all 256 FN episodes and the pre- and post-SOP cohorts (Appendix A). Since patients presented to the OD only on working days and between 8 a.m. and 4 p.m., whereas patients‘ encounters in the pedER occurred at any time, a further analysis stratified per place of first encounter was not performed.

This study might reveal a new relationship between symptoms at presentation and TTA. Patients with either fever or tachycardia received more rapidly their first dose of antibiotic compared to patients with FN without these symptoms. Despite detecting a trend to a shorter TTA, no statistically significant correlation between hypotension and TTA was observed. However, the number of hypotensive patients at first encounter was low (N = 3; Table 3). It is unclear whether these observations are extendable to other medical providers.

Setting a strict time window as a goal for the administration of the first antibiotic dose in patients with FN is controversial; however, international recommendations generally agree on a 60 min target [4,14,15]. While some studies reported that shortening TTA reduced the rate of complications, admission to ICU and overall mortality [3,6,16,17], other studies failed to confirm those benefits [18,19]. Our retrospective trial did not show any correlation between TTA and rate of adverse events (Table 4). Comparable to previous studies [20,21], we observed a higher rate of sepsis and bacteremia occurring in patients with leukemia and lymphoma compared to solid tumors (Appendix A and Appendix A). The lack of a significant difference in outcome between the pre-SOP and post-SOP cohort stratified per disease groups might be related to the modest number of FN episodes collected in this study [22,23]. In a subgroup analysis, patients with hematologic malignancies receiving vasopressors had a longer TTA (*p* = 0.025, respectively). However, data should be interpreted with caution because of the low number of AEs (N = 4). In contrast, patients with a solid tumor developing sepsis (N = 7) showed a shorter TTA (*p* = 0.046, respectively) (Appendix A and Appendix A). Peyroney et al. included adult patients with FN presenting to the outpatient department in their prospective study evaluating the effect of TTA on clinical outcome. Patients experiencing admission to ICU or death received more rapidly the first antibiotic dose and TTA was not associated with the outcome [24]. However, the authors provided no information regarding the general condition and symptoms at time of presentation. In our cohort, patients with FN presenting with a temperature of >38 °C and/or tachycardia, had a reduction in TTA. Therefore, symptoms play a role as a confounding factor when interpreting the results regarding AEs or outcomes [25]. Recently, Koenig et al. stated that the TTA was related to the clinical outcome only in patients presenting with severe disease (defined as severe sepsis or reduced clinical condition) and that those patients without severe disease do not benefit from shorter TTA [9]. In a more recent systematic literature review, Simon et al. showed that the 60 min target improves outcome in FN patients presenting with sepsis or septic shock but might be inadequate for well-appearing pediatric cancer patients with fever [26]. In a subgroup analysis of the patients in our study with symptoms at presentation (tachycardia AND/OR fever AND/OR hypotension), no change in adverse events was detectable, when a shorter TTA was achieved.

Our study may have failed to detect an association between TTA and clinical outcomes for several reasons. First, conceptual differences in the definition of TTA complicate comparisons with previous studies. Some authors define TTA as the interval between the onset of fever and antibiotic administration [3], whereas others define it as the interval between hospital arrival and antibiotic infusion [10]. Consequently, studies using the former definition may not account for geographical or logistical barriers that can delay transport to the hospital and prolong the time from symptom onset to treatment. Second, differences in patient management during, and between, chemotherapy cycles may also contribute to discrepancies between studies. Some countries recommend antibiotic prophylaxis in specific settings [27], a practice not adopted in our cohort. Although there is substantial evidence supporting the critical role of prompt antibiotic administration in preventing complications of febrile neutropenia [3,4,9,10], the factors outlined above, combined with our relatively small sample size and the paucity of serious adverse events in FN episodes of pediatric cancer patients at all, likely limited our ability to detect a significant effect in this study.

Previous studies suggested a relationship between several laboratory parameters and the risk to develop AEs in FN episodes. Different levels of neutropenia were linked to a higher rate of bacteremia (ANC < 100/µL) [28], sepsis (ANC < 300/µL) [29] and more generally severe complications (ANC < 500/µL) [30,31]. Despite observing significant differences in the neutrophil counts of patients with sepsis and bacteremia compared to those without AEs, the present investigation was not able to define any increased risk for AEs in patients with a particularly low ANC at presentation (ANC < 200/µL). Low platelet counts were associated with the risk of AEs in our study. Similar results have been reported [26,28]. CRP values were suggested as prognostic marker to predict the course of FN [29,30]. We observed that patients with an initial CRP value > 50 mg/L (N = 45) had a nearly 3-fold higher risk of requiring respiratory support. Similarly to other studies, a CRP value > 90 mg/L was associated with an overall increased risk of AEs (Appendix A) [30,31]. In addition, of the 38 patients characterized by a maximal CRP > 90 mg/L, 28 had a LOS of at least a week and an overall median LOS of 9 days. The only fatal event recorded in this study was within this subgroup.

This study presented several shortcomings inherent to its retrospective design. Investigators had limited control over data collection. Documentation of the time of admission and administration of antibiotics might have been inaccurate, since the median TTA in our institution was longer than the TTA generally reported in other medical providers, even though mortality rates and outcomes were comparable or lower [7]. The sample size was definitely too low to appreciate differences in mortality or occurrence of adverse events, given the generally very low rate of those in pediatric patients with FN. Nonetheless, the cohort size of the present investigation was comparable to four previously published, peer-reviewed studies able to assess differences in outcome [8,10,18,32]. Thereby, additional factors (e.g., stage of cancer, any aggravating comorbidities, types of indwelling catheter, differences between outpatient and emergency settings or distance of patients’ residence from the medical provider) probably played a role in determining outcome.

The priority of the SOP presented in this study was to expedite antibiotic administration. Though unlikely, this might result in administration of antibiotics to patients who retrospectively did not require it. Exposure to antibiotics could cause multiple side effects from neglectable to potentially lethal ones, such as anaphylaxis. In our cohort, 38 of 265 episodes of FN retrospectively were considered as being no FN episode (see Figure 1), but patients received antibiotics. Notably, even patients with fever without neutropenia making up the largest proportion of the 38 episodes may receive an antibiotic regimen similar to FN in our hospital.

Additionally, educating key stakeholders is critical for developing procedural pathways and for enhancing their effectiveness. Regular training sessions for hospital personnel, together with measures to streamline procedural pathways (e.g., maintaining an emergency diagnostic set for FN patients readily available in both OD and pedER or inclusion of emergency warnings in the electronic patient charts for pediatric oncology patients with fever), could improve TTA. Moreover, stakeholders not only include physicians and nursing staff but also, importantly, caregivers and patients themselves. Therefore, it is essential to strengthen the education of patients and their caregivers through multiple channels (e.g., posters about FN in the waiting room or inclusion of emergency warnings in patient’s existing health apps). Such strategies may increase vigilance and enable prompt recognition of and response to FN.

## 5. Conclusions

Overall, the current study evaluated a simple and feasible method to reduce the TTA in pediatric cancer patients with FN in the outpatient department. The TTA improvement, however, did not correspond to a comparable reduction in AEs in the course of FN episodes in our cohort. Many reasons could have played a role in the lack of impact on the overall outcome in our observation including the retrospective design of the study itself. Prospective evaluations of standardized procedures may further improve guidelines for the treatment of FN and should determine generalizability of this SOP across other pediatric oncology settings. Ultimately, FN remains one of the most frequent and potentially lethal complications experienced by children with cancer and further insight is required to ameliorate procedural pathways to improve the treatment of patients with FN.

## Figures and Tables

**Figure 1 cancers-17-03280-f001:**
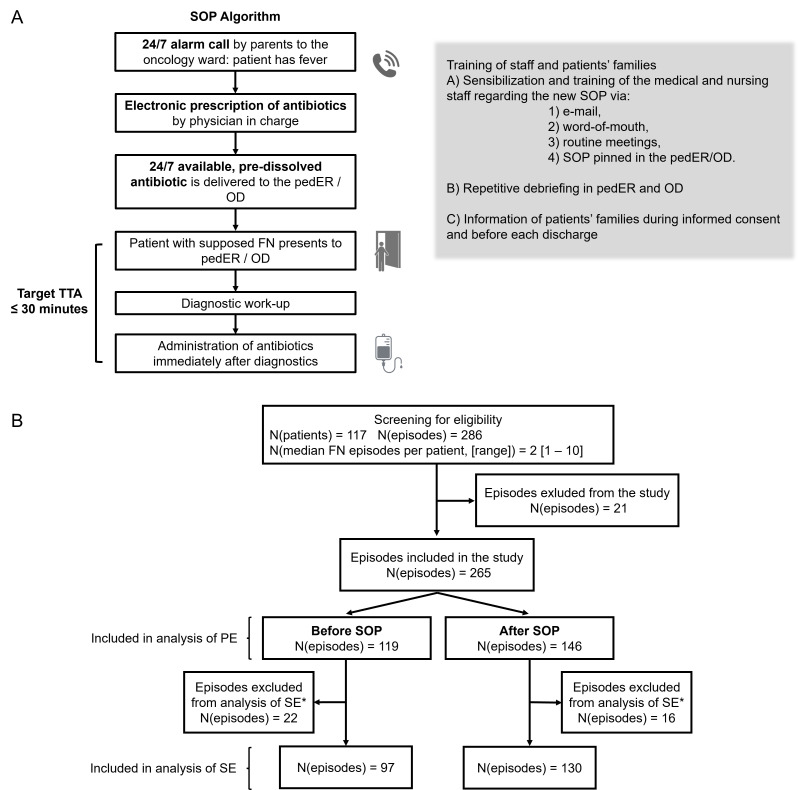
SOP algorithm and patient selection. (**A**) Flow chart representing the procedural algorithm of the new SOP. (**B**) Flow diagram of patient inclusion. FN, febrile neutropenia; N, number; N(episodes), number of FN episodes; N(patients), number of patients; OD, pediatric oncology outpatient department; pedER, pediatric emergency department; PE, primary endpoint; SE, secondary endpoint; SOP, standard of procedure; TTA, time-to-antibiotic administration. * Based on their medical history and clinical presentation, these patients were initially admitted and treated for FN, as it was the most likely diagnosis at presentation—however, FN was ruled out a posteriori and the patients had either fever without neutropenia or other conditions.

**Figure 2 cancers-17-03280-f002:**
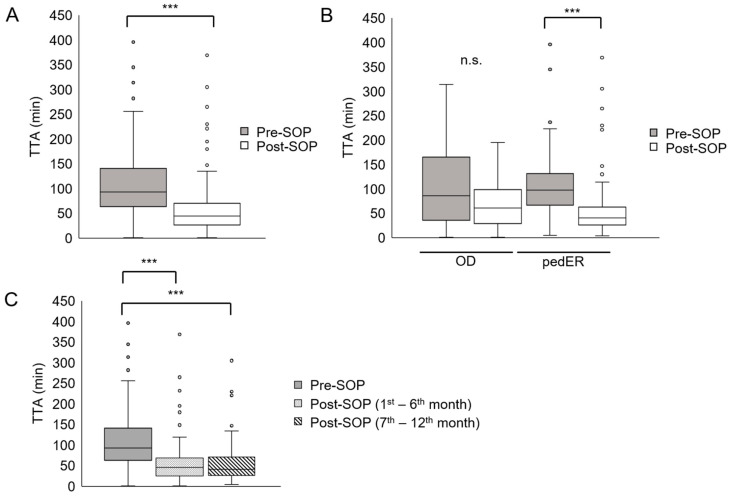
Reduction in TTA after adaption of SOP. (**A**) TTA in the pre-SOP (grey box) and in the post-SOP cohort (white box). There is a significant relationship between the two variables (adaptation of SOP and TTA). FN episodes treated after adapting the SOP are more likely to have a shorter TTA (*** *p* < 0.001). As-treated analysis considering pre-SOP N = 119 and post-SOP N = 146. Pre-SOP median (IQR) = 93 min (64–141); post-SOP median (IQR) = 44 min (27–70). (**B**) TTA before and after adapting the SOP (respectively, gray and white boxes) both in OD (pediatric oncology outpatient department, far-left and central-left box) and in pedER (pediatric emergency room, far-right and central-right box). OD pre-SOP median (IQR) = 86.5 min (38.3–158.3); OD post-SOP median (IQR) = 60.5 min (33–97)). pedER pre-SOP median (IQR) = 98 min (67–129); pedER post-SOP median (IQR) = 40.5 min (26.8–52.3). *** *p* < 0.001; n.s., not significant. (**C**) Reduction in median TTA persists over time after introduction of SOP. Pre-SOP median (IQR) = 93 min (64–141); post-SOP (1st–6th month) median (IQR) = 46 min (25–65); post-SOP (7th–12th month) median (IQR) = 42 min (27–70); *** *p* < 0.001.

**Table 1 cancers-17-03280-t001:** Patient characteristics.

Variable	N	%	N	%
	Pre-SOP		Post-SOP	
Patients	47		70	
FN episodes ^§^	119	44.9	146	55.1
Age, years (median; range)	11.1; 0.8–18.9		9.2; 0.5–18.6	
Gender ^§§^				
Male *	75	63.0	56	38.4
Female *	44	37.0	90	61.6
Diagnosis ^§§^				
Hematologic malignancy	59	49.6	73	50.0
ALL	36	30.3	40	27.4
AML	7	5.9	12	8.2
Lymphoma	14	11.8	13	8.9
Others	2	1.7	8	5.5
Solid tumor	60	50.4	73	50.0
Ewing	16	13.4	19	13.0
Germ cell tumor	5	4.2	1	0.7
Nephroblastoma	8	6.7	4	2.7
Neuroblastoma	12	10.1	14	9.6
Osteosarcoma	13	10.9	3	2.1
CNS tumor	1	0.7	4	2.7
Others	5	4.2	28	19.2

AML, acute myeloid leukemia; ALL, acute lymphoblastic leukemia; N, number; SD, standard deviation. ^§^ Percentages are calculated relative to the total number of FN episodes (N = 265). ^§§^ Percentages are calculated relative to the number of FN episodes in each cohort, 119 and 146, respectively. * *p*-value < 0.01.

**Table 2 cancers-17-03280-t002:** Stratified analysis of time-to-antibiotics in OD and pedER.

All	Pre-SOP		Post-SOP		
	N	%	N	%	
TTA ≤ 30 min	7 (119) *	5.9	48 (146)	32.9	*p* < 0.0001
TTA 31–60 min	20 (119)	16.8	48 (146)	32.9	*p* < 0.01
TTA 61–90 min	29 (119)	24.4	23 (146)	15.8	n.s.
TTA 91–120 min	25 (119)	21.0	12 (146)	8.2	*p* < 0.01
TTA > 120 min	38 (119)	31.9	15 (146)	10.3	*p* < 0.0001
TTA ≤ 60 min	27 (119)	22.7	96 (146)	65.8	*p* < 0.0001
**OD**					
TTA ≤ 30 min	3 (26)	11.5	9 (38)	23.7	n.s.
TTA 31–60 min	7 (26)	26.9	10 (38)	26.3	n.s.
TTA 61–90 min	5 (26)	19.2	8 (38)	21.1	n.s.
TTA 91–120 min	2 (26)	7.7	7 (38)	18.4	n.s.
TTA > 120 min	9 (26)	34.6	4 (38)	10.5	*p* < 0.01
TTA ≤ 60 min	10 (26)	38.4	19 (38)	50.0	n.s.
**pedER**					
TTA ≤ 30 min	4 (93)	4.3	39 (108)	36.1	*p* < 0.0001
TTA 31–60 min	13 (93)	14.0	41 (108)	35.2	*p* < 0.001
TTA 61–90 min	25 (93)	25.8	12 (108)	13.9	*p* < 0.01
TTA 91–120 min	22 (93)	24.7	5 (108)	4.6	*p* < 0.0001
TTA > 120 min	29 (93)	31.2	11 (108)	10.2	*p* < 0.001
TTA ≤ 60 min	17 (93)	17.3	80 (108)	74.1	*p* < 0.0001

N, number of FN episodes; n.s., not significant; OD, pediatric oncology outpatient department; pedER, pediatric emergency room. TTA, time-to-antibiotics. * The number of FN episodes used for the calculation of the percentages is displayed in brackets for each subgroup.

**Table 3 cancers-17-03280-t003:** Influence of findings at presentation on TTA.

	Findings	N (%)	TTA (min), Median (IQR)	*p*-Value
Tachycardia	present	114 (43.0)	52 (30–96)	*p* < 0.01
	absent	151 (57.0)	75 (41.5–118)	
Hypotension	present	3 (1.1)	50 (39.5–62.5)	n.s.
	absent	262 (98.9)	61 (33.75–109.25)	
Temperature (°C)	<38°	90 (34.0)	81 (45–127)	
	38.0–38.9°	139 (52.5)	61 (32–100)	*p* < 0.05
	≥39.0°	36 (13.5)	45 (28–86)	

Vitals were taken by nurses at presentation to the hospital. Some patients showed no fever because either temperature decreased spontaneously or their parents administered antipyretics before arrival. Percentages were calculated relative to the total number of FN episodes (N = 265). Tachycardia and hypotension were defined according to national age-adjusted reference values [11]. IQR, interquartile range; min, minutes; N, number of FN episodes; n.s., not significant.

**Table 4 cancers-17-03280-t004:** Length of stay and adverse events during FN episodes.

	Pre-SOP (N = 97)		Post-SOP (N = 130)	
	Days		Days	
LOS (days), median (IQR)	6 (4–8)		7 (4–10)	
	N	%	N	%
Sepsis	13	13.4	12	9.2
ICU admission	3	3.1	5	3.8
Vasopressors/inotropic agents	4	4.1	4	3.1
Respiratory support	8	8.2	7	5.4
Composite AEs^§^	17	17.5	16	12.3
Mortality	1	1.0	0	0

No statistically significant difference between any of the parameters. § Composite adverse events (AEs) are defined as the occurrence of at least one of the followings: sepsis, admission to ICU, use of vasopressors, need of respiratory support or death. ICU, intensive care unit; IQR, interquartile range; LOS, length of stay; N, number of FN episodes.

## Data Availability

Datasets on which the conclusions of the paper rely are stored according to local data security policy at the Medical Center of the University of Freiburg. The datasets generated and/or analyzed during the current study are available from the corresponding author on reasonable request.

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
