# Peer review of "Reducing the Time-to-Antibiotic by Adapting a Standard of Procedure for the Treatment of Pediatric Cancer Patients Presenting with Febrile Neutropenia—A Comparative Analysis of Two Patient Cohorts"

_cancers, 2025, doi:10.3390/cancers17203280_

Round 1

Reviewer 1 Report

Comments and Suggestions for Authors

This is a well written report that addresses an important issue in pediatric oncology. The authors highlight the ability of clearly defined clinical pathways to improve important measures in care delivery. They address the challenges in demonstrating the ability of these pathways to prevent rare events, despite the likelihood that they in fact do improve care overall. While several reports highlighting efforts to improve TTA exist, this remains a relevant topic in pediatric oncology.

There are a few areas discussed below that would strengthen this manuscript.

Key areas to consider:

  • Were patients with and without central lines included in this pathway? Did type of central catheters used affect TTA or likelihood of blood stream infection?
  • Were there any differences in antibiotic prophylaxis approaches during chemo cycles between the two cohorts?
  • Antibiotics used in this pathway are described in the supplementals but would be worth discussing in the manuscript so that others can more readily consider implementation at their institution
  • In developing pathways of this nature, education of key stakeholders is critical. Would consider elaborating on who stakeholders were and what the next steps will be to continue to improve outcomes
  • Balancing measures: Please consider any balance measures that may be relevant to this work. For example, did the focus on shortening time to administration, though perhaps unlikely, result in administration of antibiotic to patients who in retrospect did not meet fever criteria, had other contraindication, etc? Did pre-dilution of antibiotic substantially increase cost of care, result in wasted doses, etc?
  • Please clarify this sentence or elaborate “A typical example would be a family informing the medical staff that their child had FN only several minutes after encounter and not in advance, because to them it had not appeared impaired yet.”
  • Minor edits:
    • Line 229: “and that those patients without severe disease do not profit benefit from shorter TTA”
    • Line 300: “showed that the 60-minute target 300 improves outcome in FIN FN patients presenting with sepsis or septic shock”

Reviewer 2 Report

Comments and Suggestions for Authors

The manuscript entitled “Reducing the time-to-antibiotic by adapting a standard of procedure for the treatment of pediatric cancer patients presenting with febrile neutropenia”, is a retrospective, single-center, cohort study, which intends to demonstrate the possibility of timely controlling febrile neutropenia (FN), a major side-effect of chemotherapy, in various cancers, mainly, hematological malignancies. FN is a bacterial infection caused in the bloodstream of the cancer patients undergoing chemotherapy. FN is an emergency condition, which demands immediate medical attention. In the current study, the patients include a range of pediatric cancers, majority of whom are depended on their respective caregiver. To counter FN, it is critical to take into consideration time-to-antibiotics (TTA), as early as possible. In the above context, the current manuscript may be considered to have significant importance in clinical settings that frequently encounters major adverse effects on patients undergoing chemotherapy.

As per the study design, the authors have divided the patients randomly into two groups, namely Pre-SOP and Post-SOP. Pre-SOP is the standard protocols followed in clinical setup wherein onset of symptom of FN, is followed by diagnosis, and subsequent antibiotic treatment based on the diagnostic results. As per the authors, all of these requires a median time of approx. 90 minutes TTA, which is a very narrow window, and may result in near mortality condition. Hence, it is critical to lower the TTA significantly, thereby widening the window of treatment for timely response to FN. The post-SOP procedure included an upper time limit of less than 30 minutes post onset of FN symptoms, to lower the probability of mortality because of FN.

In this context, some conceptual and technical commentaries have been made below:

  1. In the methodology section, the authors have clearly mentioned a clear time limit of maximally 30 minutes between admission to the pedER or OD and start of the administration of antibiotics, even without waiting for the absolute neutrophil count (ANC). Yet, we are unable to understand the data in figure 2 and table 2, wherein post-SOP TTA data ranges way above 30 minutes, and instead are in terms of median 44 minutes.
  2. As per line 96-102, the primary endpoint (PE) includes the median TTA, while secondary endpoints (SE) include occurrence of adverse effects. As per the figure 1 data, PE includes 38 episodes [22+16], while SE includes 227 episodes [[97+130]. Assumingly that the figure 2 and table 2 are data of PE, why have the number of episodes not been 22 and 16 for pre- and post SOP respectively, instead it is 119 and 146 (which probably includes both SE and PE)
  3. A typo was identified in line 301, FIN/ FN patient?
  4. The title of the manuscript may require a minor modification in order to distinguish between the pre-SOP and post-SOP.
  5. Overall, the manuscript has been well written, with adequate literature background. The conclusion will be decided upon the response of the authors.

Based on the response to the above commentaries, we will take a call on the outcome of the manuscript.

Reviewer 3 Report

Comments and Suggestions for Authors

The article “Reducing the time-to-antibiotic by adapting a standard of procedure for the treatment of pediatric cancer patients presenting with febrile neutropenia” is interesting, I have following comments/suggestions,

  1. Authors have included febrile neutropenia patients with poor clinical condition, and this needs clearer justification.
  2. Authors have not clearly stated the process of SOP implementation, and more detail is required that how the uniformity was ensured.
  3. No explanation is given for the calculated sample size.
  4. Statistical analysis is not appropriate as regression models adjusting for confounders can provide more rational results
  5. Limitations should also mention confounding factors such as cancer stage, comorbidities, and differences between outpatient and emergency settings.

Reviewer 4 Report

Comments and Suggestions for Authors

The research aims to determine if decreasing the time-to-antibiotic (TTA) can improve outcomes in pediatric febrile neutropenia. Despite the reduction of TTA from 93 to 44 minutes, there was no significant difference observed in adverse events (AEs). Furthermore, this contradicts the foundational assumption of the study regarding the benefits of a shorter TTA, highlighting the need for more precise clarification.  
The revised SOP established a target of administering antibiotics within 30 minutes; however, only 32.9% of patients achieved this benchmark following the implementation of the SOP, indicating that the majority of patients fell short of the goal. The manuscript highlights the SOP as a success; however, this assertion conflicts with the study's main performance indicator. The generalizability of the study across clinical settings surpasses the existing evidence, and it is essential to more clearly recognize the limitations in OD performance. Patients with severe symptoms had shorter TTAs; nonetheless, outcomes did not show improvement within these groups. The body of literature actually presents various studies suggesting improved results with prompt antibiotic delivery; nonetheless, the conversation ultimately concludes that there is no demonstrable benefit and raises questions about the predictive value of TTA. A more comprehensive integration of how this study aligns with the broader evidence would enhance the narrative significantly.

Round 2

Reviewer 2 Report

Comments and Suggestions for Authors

The authors have satisfactorily responded to the queries raised. Substantial changes have also been made in the manuscript as suggested to improvise the manuscript. The manuscript may be proceeded to the publication stage. 

Reviewer 3 Report

Comments and Suggestions for Authors

Authors have sddressed my comments/suggestions in their revised submission. 

Reviewer 4 Report

Comments and Suggestions for Authors

The revisions to the article effectively addressed the reviewer's concerns. The authors reported that, despite the reduction in time-to-antibiotic (TTA) from 93 to 44 minutes, there was no increase in adverse outcomes. The discussion highlights this contrast and connects the absence of observable outcome improvements to factors such as the retrospective design, small sample size, limited severe cases, and the exacerbating effects of symptoms like fever and tachycardia. Recent research indicates that a shorter TTA is most advantageous for individuals with severe illness. Their results are contextualized within existing research.

Moreover, the authors corrected their previous error regarding the use of the standard operating procedure (SOP). They report that only 32.9% of patients achieved the 30-minute goal, indicating significant, though incomplete, improvements from the intervention. The new statement emphasizes the necessity for additional measures to enhance conditions, including public education and reforming current practices. The article discusses the distinctions between the emergency room and the outpatient department. The changes were particularly evident in the emergency room due to issues with the building and the staff.

The discussion concludes by evaluating the generalizability and its limitations of this study, which encompass its retrospective design, concerns about record accuracy, a small sample size, and the risk of antibiotic overuse. The revised version includes several comparative studies demonstrating how its conclusions align with the varied data on TTA outcomes. The update shifts the study from merely demonstrating effectiveness to highlighting improvements in the quality of care. Clear and balanced reasoning, which addresses the reviewer's main points of criticism, achieves this goal.

Comments on the Quality of English Language

The quality of English in the revised manuscript is generally clear, formal, and scientifically appropriate.